# Development of an Allergic Rhinitis Diagnosis Application Using the Total Tear IgE Detection Kit for Examining Nasal Fluid: Comparison and Combination with the Conventional Nasal Smear Examination for Eosinophils

**Hiroshi Kumanomidou** [1,2,*] **and Mitsuhiro Okano** [2]

1   Kumanomidou ENT Clinic, HNB #1F, 2-1-3, Kamitakada, Nakano-ku, Tokyo 164-0002, Japan
2   Department of Otorhinolaryngology, Graduate School of Medicine, International University of Health and Welfare, 4-3 Kozunomiri, Narita-shi 286-8686, Japan
*   Correspondence: hkumanom-ent@am.wakwak.com; Tel.: +81-3-5345-6201

**Abstract:** Allergic rhinitis (AR) is a type I allergic disease characterized by immunoglobulin E (IgE) -mediated hypersensitivity of the nasal mucosa. Here, we focused on a commercial test kit named Allerwatch® (AW) for the diagnosis of allergic conjunctivitis (AC) in which total tear IgE is qualitatively detected based on immunochromatography. We evaluated the usefulness of the AW test for detecting total IgE in the nasal discharge of AR and non-allergic rhinitis (non-AR) patients in comparison and combination with the conventional nasal smear examination for eosinophils. Using the AW test, total IgE in nasal fluid was detected in 64.76% of the AR patients and 11.11% of the non-AR patients, with a significant difference between the groups ($p < 0.001$). As compared to non-AR, the sensitivity and specificity of the detection of total IgE in nasal fluid for detecting AR were 64.76% and 88.89%, respectively. In the AR patients, house dust mites (57.1% of patients) and Japanese cedar pollen (93.3% of patients) were the major sensitizing antigens. When we considered a positive result in either of the two examinations to indicate a positive result, the rate of positivity in AR patients increased to 78.10%. As compared to non-AR, the sensitivity and specificity of the combination of both examinations for detecting AR were 78.10% and 83.33%, respectively. The AW test in the nasal cavity and the qualitative measurement of total IgE in nasal fluid may enable the detection of allergic elements in patients who present to a medical institution with nasal symptoms. In addition, the detection rate is increased when combined with the nasal smear examination for eosinophils.

**Keywords:** allergic rhinitis; total tear IgE; allergic conjunctivitis; immunochromatography; rapid diagnostic test tool

## 1. Introduction

Allergic rhinitis (AR) is a type I allergic disease characterized by immunoglobulin E (IgE)-mediated hypersensitivity of the nasal mucosa. The diagnosis of AR is based on the allergic elements and sensitization to inhalant allergens, and examinations for the presence of allergic elements in AR include the examination of serum total IgE level, in blood eosinophilic granulocyte and eosinophils in nasal smear [1]. However, these examinations have limitations for rapid and sensitive diagnosis [2]. For example, the measurement of serum total IgE offers conflicting outcomes and divergent opinions in relation to symptom severity and Karli et al. suggested that serum total IgE is a factor that can be used for confirming the diagnosis, but its routine use is not recommended due to the high cost and long testing time [3,4]. The measurement of total tear IgE is useful for the diagnosis of allergic elements in allergic conjunctivitis (AC), which is another typical type I allergic disease that is often comorbid with AR. Total tear IgE can be measured in the clinical setting using a commercially available kit named Allerwatch® (AW) made in Japan [5]. This AW

test shows high sensitivity and specificity for diagnosing AC rapidly [6–9]. A previous study reported that a positive result for IgE in nasal fluid was obtained with the AW test in 14 out of 22 patients with Japanese cedar pollinosis using an artificial allergen exposure chamber out of pollen season (sensitivity: 63.6%) [10]. The AW test was improved in 2013; the compression of the cellulose nonwoven fabric and the lamination of the specimen collection area were improved to shorten the specimen absorption time, and to enable testing with a smaller specimen volume [11]. Because little is known about whether the detection of total IgE in nasal discharge is useful for the diagnosis of allergic elements in AR, we evaluated the usefulness of the improved AW test for detecting IgE in nasal discharge in comparison and combination with the conventional nasal smear examination for eosinophils. The results presented here may provide a basis for developing a new, simple, and rapid diagnostic method for allergic elements in AR.

## 2. Materials and Methods

### 2.1. Subjects

This study was an analytical observational study conducted at the Kumanomidou ENT Clinic in Tokyo, Japan, between March 2019 and October 2021. In total, 105 AR patients (20 to 80 years of age; mean age: 52.48 years; 54 males and 51 females) who were diagnosed with paroxysmal nasal symptoms, such as sneezing, rhinorrhea, and nasal congestion and showed sensitization to inhaled allergens, such as house dust mites, in which the onset pattern was consistent with sensitization were included. As controls, 18 non-allergic rhinitis (non-AR) patients who showed paroxysmal nasal symptoms, but were negative for serum antigen-specific IgE against house dust mites, Japanese cedar pollen, Japanese cypress pollen, and other allergens related to the onset of the symptoms, such as animal dander. In selecting patients with AR patients and controls (non-AR patients), those with acute inflammation with fever and obvious purulent nasal discharge and patients with chronic sinusitis with severe nasal polyps were excluded. AR patients were excluded if they were on sublingual immunotherapy.

This study was conducted in accordance with the 1975 Declaration of Helsinki and its subsequent amendments, and was approved by the Institutional Review Board of the International University of Health and Welfare Graduate School of Medicine (18-Im-012. 20-Im-004). Written informed consent was provided by each subject for inclusion in the study.

### 2.2. Detection of Total IgE in Nasal Fluid

The AW test kit strips were inserted into the nasal cavity to absorb nasal discharge, and the strip was removed after confirming that it was moistened. The strip was then immersed in a 2.0 mL Eppendorf tube with a few drops of development solution buffer for 10 min at room temperature. The strip contains reagents necessary for the measurement of total IgE level, and a red line develops at the specified position of the strip when an IgE antibody is present. According to the manufacturer's instructions, the test results are categorized into three grades: Grade 0, no detectable IgE (no line); Grade 1, a low total IgE level (test line weaker than the control); and Grade 2, a high total IgE level (test line similar to or stronger than the control) [5,8,9]. In this study, Grades 1 and 2 were taken to indicate a positive result for the presence of total IgE level in nasal fluid (Figure 1).

### 2.3. Nasal Smear Examination for Eosinophils

After collecting nasal discharge by gently rubbing the surface of the inferior turbinate with a cotton swab, the nasal discharge was applied to a glass slide, and a nasal smear was prepared. Eosinophils were evaluated as follows: none (−); visible under high (400×) magnification (+); visible cluster formation (+++); and between + and +++ (++). In this study, detection of (+) or greater was considered positive [2].

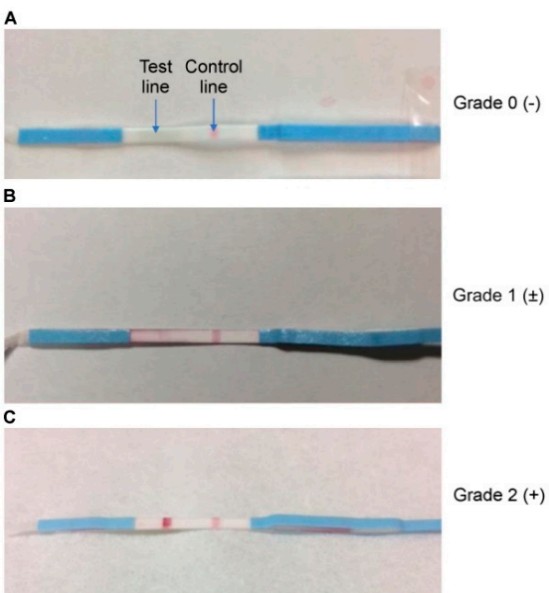

**Figure 1.** Detection of total IgE in nasal fluid using the AW test: (**A**): Grade 0. (**B**): Grade 1. (**C**): Grade 2.

*2.4. Severity of Symptoms and Quality of Life*

Using the Japanese Rhinoconjunctivitis Quality of Life Questionnaire (JRQLQ No. 1), patients were asked to rate their recent (1 to 2 weeks) naso-ocular symptoms, life limitations, and global status on a 5-point scale from 0 to 4 [1].

*2.5. Statistical Analysis*

Values are expressed as the mean of each subject group. The *t*-test and chi-square test were used to compare the data between groups. Statistical analyses were performed using GraphPad Prism (GraphPad Software, Inc., La Jolla, CA, USA) version 8, and *p* values < 0.05 were considered to indicate statistical significance.

## 3. Results

*3.1. Detection Rate of Total IgE in Nasal Fluid*

With the AW test, total IgE in nasal fluid was detected in 68 out of 105 (64.76%) AR patients, and in 2 out of 18 (11.11%) non-AR patients, with a significant difference between the two groups ($p < 0.001$ by the chi-square test; Figure 2A). As compared to non-AR, the sensitivity and specificity of the detection of total IgE in nasal fluid for detecting AR were 64.76% and 88.89%, respectively.

*3.2. Relationship between the Detection Rates of Total IgE in Nasal Fluid and Eosinophils in Nasal Smear*

Eosinophils were detected in the nasal smear of 52 out of 105 (49.52%) AR patients, and in 1 out of 18 (5.56%) non-AR patients, with a significant difference between the two groups ($p < 0.001$; Figure 2B). Among the AR patients, the detection rate of total IgE in nasal fluid was higher than that of eosinophils in nasal smear; however, the difference was not statistically significant ($p = 0.077$). Among patients with AR, 38 out of 105 (36.19%) patients were positive for both total IgE in nasal fluid and eosinophils in nasal smear, 30 (28.57%) patients were positive only for total IgE in nasal fluid, 14 (13.33%) patients were positive only for eosinophils in nasal smear, and 23 (21.90%) patients were negative for both (Table 1). In contrast, none of the eighteen non-AR patients were positive for both total IgE in nasal fluid and eosinophils in nasal smear, two (11.11%) patients were positive only for total IgE in nasal fluid, one (5.56%) patient was positive only for eosinophils in nasal smear, and fifteen (83.33%) patients were negative for both (Table 2). When we considered a positive result in either of the two tests to indicate a positive result, the positivity rate in AR patients

increased to 78.10% (Figure 2C). As compared to non-AR, the sensitivity and specificity of the combination of both tests for detecting AR were 78.10% and 83.33%, respectively.

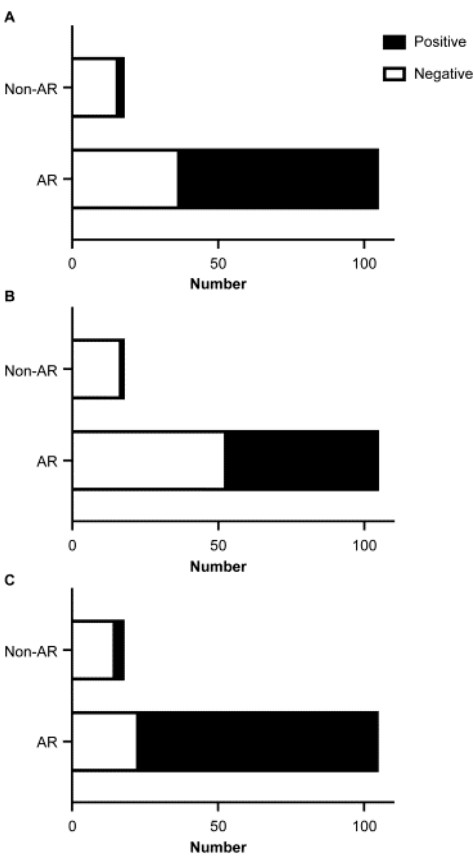

**Figure 2.** Comparison of the detection rates of total IgE in nasal fluid using the AW test and the conventional nasal smear examination for eosinophils between the AR and non-AR patients: (**A**): Total IgE in nasal fluid. (**B**): Eosinophils in nasal smear. (**C**): Combination of the two examinations.

**Table 1.** Comparison of the sensitivity and specificity for the diagnosis of allergic rhinitis (AR) between the detection of total IgE in nasal fluid using the AW test and the conventional nasal smear examination for eosinophils.

|  |  | Nasal Smear Examination for Eosinophils | |
| --- | --- | --- | --- |
|  |  | **Positive** | **Negative** |
| detection of total IgE in nasal | Positive | 38 | 30 |
| fluid using the AW test | Negative | 14 | 23 |

**Table 2.** Comparison of the sensitivity and specificity for the diagnosis of non-allergic rhinitis (non-AR) between the detection of total IgE in nasal fluid using the AW test and the conventional nasal smear examination for eosinophils.

|  |  | Nasal Smear Examination for Eosinophils | |
| --- | --- | --- | --- |
|  |  | **Positive** | **Negative** |
| detection of total IgE in nasal | Positive | 0 | 2 |
| fluid using the AW test | Negative | 1 | 15 |

*3.3. Role of Nasal Total IgE in the Severity of Symptoms and QOL in AR*

The naso-ocular symptoms, QOL, and global status were assessed using the JRQLQ No. 1 Naso-ocular symptoms score: no significant differences in sneeze ($p$ = 0.991;

Figure 3A), rhinorrhea (*p* = 0.827; Figure 3B), nasal congestion (*p* = 0.621; Figure 3C), itchy nose (*p* = 0.786; Figure 3D), and itchy eyes (*p* = 0.985; Figure 3E) were seen between the patients who were positive and those who were negative for total IgE in nasal fluid. Severe watery eyes tended to be more common in patients who were positive for total IgE in nasal fluid (*p* = 0.092; Figure 3F). QOL scores: no significant differences in daily life (*p* = 0.958; Figure 3G), outdoor (*p* = 0.916; Figure 3H), social (*p* = 0.167; Figure 3I), sleep (*p* = 0.092; Figure 3J), body (*p* = 0.579; Figure 3K), and psycho-life (*p* = 0.577; Figure 3L). The global status as determined by a face scale was similar between the two groups (*p* = 0.951; Figure 3M).

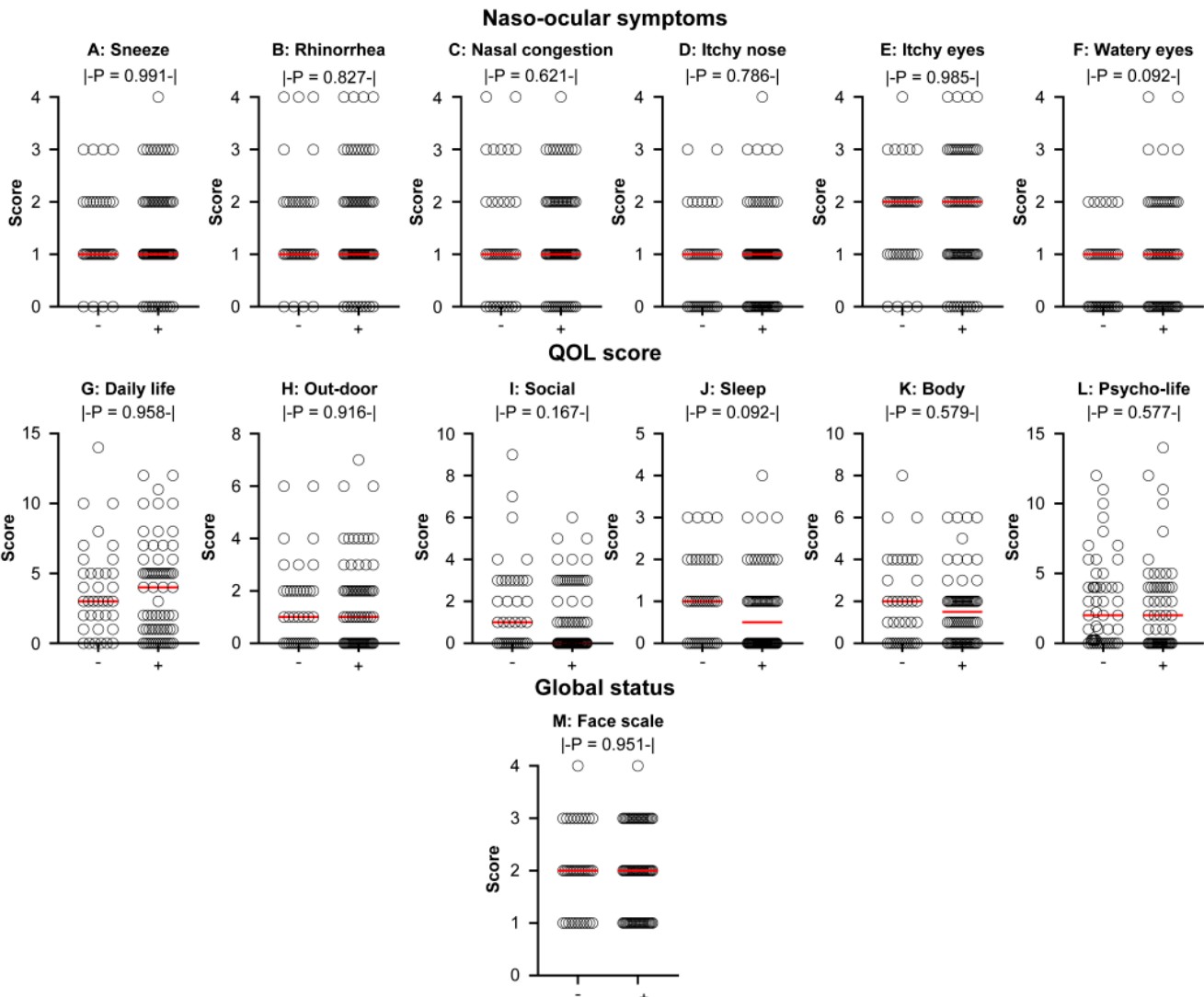

**Figure 3.** Comparison of the severity in the naso-ocular symptoms, QOL score, and global status between patients with a positive result and patients with a negative result for total IgE in nasal fluid: naso-ocular symptoms score, (**A**): Sneeze. (**B**): Rhinorrhea. (**C**): Nasal Congestion. (**D**): Itchy nose. (**E**): Itchy eyes. (**F**): Watery eyes. QOL score, (**G**): Daily life. (**H**): Outdoor. (**I**): Social. (**J**): Sleep. (**K**): Body. (**L**): Psycho-life. global status, (**M**): Global status by the face scale. *p* values were determined by the *t*-test.

*3.4. Association between Total IgE in Nasal Fluid and AR Phenotypes*

Among the 105 AR patients, 66 patients were classified as having perennial AR with or without seasonal exacerbation, and 39 patients were classified as having seasonal AR. A total of 47 of the 66 (71.2%) patients with perennial AR, and 21 of the 39 (53.8%) patients with seasonal AR were positive for total IgE in nasal fluid, which indicated that the detection

rate tended to be higher in patients with perennial AR (*p* = 0.092 by the chi-square test; Figure 4A). House dust mites (sensitization in 60 out of 105 patients; 57.1%) and Japanese cedar pollen (sensitization in 98 out of 105 patients; 93.3%) were the major sensitizing antigens. The detection rate of total IgE in nasal fluid was not significantly different in those with or without sensitization to house dust mites (*p* = 0.220 by the chi-square test; Figure 4B) or to Japanese cedar pollen (*p* = 0.695 by the chi-square test; Figure 4C).

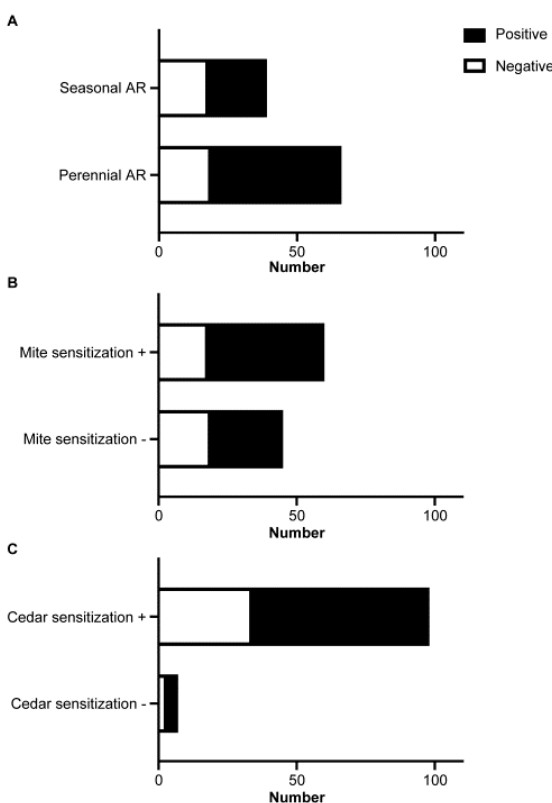

**Figure 4.** Association between the total IgE in nasal fluid by the AW test and AR phenotypes: (**A**): Seasonal versus perennial allergic rhinitis. (**B**): Sensitization to house dust mites. (**C**): Sensitization to Japanese cedar pollen.

## 4. Discussion

In the present study, total IgE in nasal fluid was detected in 64.76% of the patients with AR using the AW test. As compared to non-AR, the sensitivity and specificity of the detection of total IgE in nasal fluid for detecting AR were 64.76% and 88.89%, respectively. We believe that this is the first study to show the rate of natural exposure to allergens. A previous study published in 2011 reported a sensitivity of 63.9% and specificity of 87.5% for the detection of nasal total IgE using the AW test (before the test was improved in 2013) in patients with Japanese cedar pollinosis after artificial exposure to pollens in an environmental chamber [10]. The results of the present study are consistent with those of the previous report in terms of sensitivity and specificity, and the results were reproducible regardless of whether the patients had specific antibodies related to the AR symptoms, and whether the environment was an exposure room or the daily living environment when symptoms were present. The nasal smear examination for eosinophils is useful for detecting local allergic elements in the clinical setting. The antigen identification test includes the serum antigen-specific IgE antibody, skin reaction with presumed antigens, and nasal provocation test [1]. We planned to use nasal discharge as a specimen to detect atopic status when patients visit the clinic with nasal symptoms. In addition, this plan is useful for a comprehensive examination of atopic status, since the suspected antigen may not be available for skin reaction with presumed antigens.

The sensitivity of 64.76% is not very impressive. However, based on the definition of AR as a type I allergic disease of the nasal mucosa, it is desirable to be able to detect the IgE antibody in nasal fluid. We think that total IgE in nasal secretion may reflect atopic status. Although eosinophils and IgE antibody production are controlled by type 2 immune responses, they can be induced independently. For example, nasal eosinophils were seen in *IL-4* gene-deficient mice following antigen exposure even when total and antigen-specific IgE was not detected [12]. By combining the AW test and the nasal smear examination for eosinophils, the possibility of proving allergic elements in nasal discharge increased from 64.76% for the AW test alone and 49.52% for the nasal smear examination for eosinophils alone to 78.10% when a positive result in either test was taken to indicate a positive result. Thus, we showed that the detection rate of total IgE in nasal fluid was higher than that of eosinophils in nasal smear; although, the difference was not statistically significant. These results suggest that the detection of total IgE in nasal fluid using the AW test is better than the effect of nasal smear examination for eosinophils. We believe that this is the first study showing the usefulness of a combined examination of two local tests for improving the rapid diagnosis of AR. In this study, there was no significant association between each naso-ocular symptoms score and the positivity rate of total IgE in nasal fluid by the AW test; however, the watery eyes symptom score tended to be higher in patients with a positive AW test result. This suggests that even though the samples were collected for nasal discharge, contamination by tear fluid components flowing from the nasolacrimal duct into the nasal cavity could not be completely excluded. Among the non-AR patients with a negative serum-specific IgE antibody test result, two patients had a positive result for the AW test. As compared to AR, non-AR comprises various conditions of rhinitis, such as idiopathic rhinitis, non-allergic rhinitis with nasal eosinophilia syndrome (NARES), and local allergic rhinitis (LAR). LAR is characterized by the presence of specific IgE against the causative antigen in the nasal mucosa of patients with negative serum specific IgE antibody test and positive nasal provocation test (NPT) [13–16]. In the NPT, a filter paper disc soaked with the causative antigen is placed on the inferior turbinate mucosa, and the results are judged by the number of sneezes, nasal pruritus, amount of nasal discharge, and degree of swelling of the inferior turbinate mucosa. In Japan, only a few discs can be used (house dust and ragweed), and antihistamines and topical steroids must be discontinued before the NPT is performed, so the test is not widely used in Japanese otorhinolaryngology clinics [1]. Although no diagnostic criteria for LAR have yet been established, it is possible that these two patients suffered from LAR. The detection of total IgE in nasal fluid using the AW test may be effective in the identification of LAR; although, it needs to be combined with other examinations, and the usefulness for the diagnosis of LAR should be investigated in the future [17].

## 5. Conclusions

In conclusion, the AW test in the nasal cavity and the qualitative measurement of total IgE in nasal fluid may be useful and convenient for detecting the presence of allergic elements in patients who present to a medical institution with nasal symptoms. The results from the AW test can be a rapid test as compared with the serum antigen-specific IgE antibody test, obtained within 10 min. The detection rate is increased when combined with the conventional nasal smear examination for eosinophils. One advantage of using the AW test method for clinical decision making in the treatment of AR is that it has already been evaluated and used in the field of ophthalmology [6,9].

**Author Contributions:** Conceptualization, H.K. and M.O.; data curation, H.K.; formal analysis, H.K. and M.O.; funding acquisition, M.O.; investigation, H.K.; writing—original draft, H.K. and M.O.; writing—review and editing, M.O. All authors have read and agreed to the published version of the manuscript.

**Funding:** This research was funded by The Japan Agency for Medical Research and Development (AMED), grant number JP21ek0410089h0001.

**Institutional Review Board Statement:** The study was conducted in accordance with the Declaration of Helsinki, and approved by the Institutional Review Board of the International University of Health and Welfare Graduate School of Medicine (protocol code 18-Im-012, 12 January 2019 and 20-Im-004, 24 June 2020).

**Informed Consent Statement:** Written informed consent has been obtained from the patients to publish this paper.

**Data Availability Statement:** The data presented in this study are available on request from the corresponding author.

**Conflicts of Interest:** The authors declare no conflict of interest.

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
