# Peer review of "Development of an Allergic Rhinitis Diagnosis Application Using the Total Tear IgE Detection Kit for Examining Nasal Fluid: Comparison and Combination with the Conventional Nasal Smear Examination for Eosinophils"

_allergies, doi:10.3390/allergies2040014_

Round 1

Reviewer 1 Report

Comments and Suggestions for Authors

The authors state as the object of the report to provide a simple rapid diagnostic test for allergic rhinitis. The conclusion is that the AW test may be useful. However, the sensitivity of 65% is not very impressive. The proposed flow chart for diagnosis of allergic rhinitis is complex and not supported by any data demonstrating its usefulness. The report may best indicate that the Allerwatch (AW) test has limited usefullness in diagnosing allergic rhinitis compared with more conventional means such as history and allergen specific IgE that corresponds to the history.

Author Response

Response to Reviewer 1 Comments

Point 1:  The sensitivity of 65% is not very impressive.

Response 1: We agree with the reviewer's comment that the sensitivity of 65% is not very impressive. We added this suggestion into the revised manuscript. (Discussion p7. line 204-207)

Point 2:  The proposed flow chart for diagnosis of allergic rhinitis is complex and not supported by any data demonstrating its usefulness. 

Response 2: In accordance with the Reviewer’s comment, we have deleted the proposed flow chart. (Conclusions p8. line 256) and, we have removed the explanatory text in Fig. 5. (Conclusions p8. line 253-255)

Point 3:  The report may best indicate that the Allerwatch (AW) test has limited usefullness in diagnosing allergic rhinitis compared with more conventional means such as history and allergen specific IgE that corresponds to the history.

Response 3: We agree with the reviewer’s comment that the Allerwatch (AW) test alone has limited usefullness in diagnosing allergic rhinitis compared with more conventional means such as history and allergen specific IgE that corresponds to the history. However, AW test is the rapid test as compared with allergen specific IgE, thus it may be useful for the primary examination for allergic rhinitis.

We added the above discussion in the revised manuscript. (Discussion p7-8. line 215-220, & Conclusions p8. line 245-251)

English language and style

(x) English language and style are fine/minor spell check required

The manuscript has been carefully reviewed by an experienced editor whose first language is English and who specializes in editing papers written by scientists whose native language is not English.

Reviewer 2 Report

Comments and Suggestions for Authors

The authors present a very interesting topic - local allergic rhinitis. Difficulties in diagnostics, especially in proving the IgE-dependent mechanism of allergy, cause a long delay in the patient's qualification for the causal treatment of allergy, which is specific immunotherapy. The developed tool for the detection of locally produced IgE antibodies is a great achievement. The combination of this tool together with a nasal swab for eosinophils significantly improves diagnostics, as the authors present clearly.

The introduction is short and concise, the aim of the study is clearly defined. The methods are clearly described, as are the results obtained. The discussion is written succinctly, clearly and to the point. References are selected appropriately.

Author Response

Response to Reviewer 2 Comments

Point 1:  The introduction is short and concise, the aim of the study is clearly defined. The methods are clearly described, as are the results obtained. The discussion is written succinctly, clearly and to the point. References are selected appropriately.

Response 1: We wish to express our strong appreciation to the reviewer for these comments on our paper.

Reviewer 3 Report

Comments and Suggestions for Authors

The Article „ Development of an allergic rhinitis diagnosis application using 2 the total tear IgE detection kit for examining nasal fluid: 3 comparison and combination with the conventional nasal smear 4 examination for eosinophils” highlights the role of IgE detection in nasal fluid in the management of patients with suspicion of allergic rhinitis. I have read the paper with interest and feel that it is relevant for the area of Allergic Rhinitis management.

I suggest major revisions to improve the article and comments are made below regarding the article.

  1. In the Methods: The authors should clarify the type of patients included in the study, already diagnosed with AR (allergic rhinitis), how the diagnosis was made for AR or non-AR
  2. In the Methods: Add inclusion/ exclusion criteria and Statistical analysis
  3. In Discussion comment about localized AR
  4. References should be arranged according to the journal requests.

Author Response

Response to Reviewer 3 Comments

Point 1:   In the Methods: The authors should clarify the type of patients included in the study, already diagnosed with AR (allergic rhinitis), how the diagnosis was made for AR or non-AR

Response 1:  We thank the reviewer for this comment.

AR was diagnosed by the presence of paroxysmal nasal symptom and the presence of sensitization to inhaled allergens. Non-AR was diagnosed by the presence of paroxysmal nasal symptom without sensitization to inhaled allergens. (Subjects p2. line 63-71)

Point 2:   In the Methods: Add inclusion/ exclusion criteria and Statistical analysis

Response 2: Thank you for your suggestion.

We added the information in the revised manuscript. The inclusion/ exclusion criteria in Subject. p2. line 71-74. Statistical analysis have described in detail in. Statistical analysis p3. line 109-113.

Point 3:   In Discussion comment about localized AR

Response 3: We thank the reviewer for this comment.

We added the discussion comment in the revised manuscript. (Discussion p8. line 230-242)

Point 4:   References should be arranged according to the journal requests.

Response 4: We thank the reviewer for this comment.

We have modified the reference according to MDPI’s style for citations and reference lists.

(References p9-10. line 286-327)

English language and style

(x) Extensive editing of English language and style required

The manuscript has been carefully reviewed by an experienced editor whose first language is English and who specializes in editing papers written by scientists whose native language is not English.

Reviewer 4 Report

Comments and Suggestions for Authors

General comments:

The paper, titled as development of an allergic rhinitis diagnosis application using the total tear IgE detection kit for examining nasal fluid: comparison and combination with the conventional nasal smear examination for eosinophils, by Hiroshi Kumanomidou, Mitsuhiro Okano, to explore a new diagnosis methods with allergic rhinitis. This a simply study, and have some conclusions. I think this method is controversial.

Specific comments:

1.     The author only measured the total IgE, but did not measure the specific IgE, and total IgE could not represent the allergy.

2.     The significance of this study may be effective in the identification of local allergic rhinitis. Of course, it needs to be combined with other examinations.

3.     This check is only preliminary, and it seems that it is not better than the effect of nasal smear examination for eosinophils.

Author Response

Response to Reviewer 4 Comments

Point 1:  The author only measured the total IgE, but did not measure the specific IgE, and total IgE could not represent the allergy.

Response 1: We agree with the reviewer’s comment that total IgE could not represent the allergy to inhaled antigens. We think that total IgE in nasal secretion may reflect atopic status.

We added the above information in the revised manuscript. (Discussion p7. line 204-207)

Point 2: The significance of this study may be effective in the identification of local allergic rhinitis. Of course, it needs to be combined with other examinations.

Response 2: Thank you for your suggestion. We added the following sentences in the revised discussion:

The detection of total IgE in nasal fluid using the AW test may be effective in the identification of LAR although it needs to be combined with other examinations, the usefulness for the diagnosis of LAR should be investigated in the future . (Discussion p8. line 239-242)

Point 3: This check is only preliminary, and it seems that it is not better than the effect of nasal smear examination for eosinophils.

Response 3: We agree with the reviewer’s comment that this check is only preliminary. However, we showed in the original manuscript, that the detection rate of total IgE in nasal fluid was higher than that of eosinophils in nasal smear although the difference was not statistically significant. These results suggest that the nasal AW test is better than the effect of nasal smear examination for eosinophils.

We added the above discussion in the revised manuscript. (Discussion p7-8. line 215-220)

English language and style
(x) Moderate English changes required

The manuscript has been carefully reviewed by an experienced editor whose first language is English and who specializes in editing papers written by scientists whose native language is not English.

Round 2

Reviewer 1 Report

Comments and Suggestions for Authors

The authors conclude that the AW test may be useful. Does it really provide more useful information that just the history of symptoms consistent with allergic rhinitis. Skin testing for suspected inhalant allergens identifies allergen specific IgE rapidly and is the more usual means of evaluation. What, if any, then is there any reason to use the AW test since it provides less useful information than allergy skin tests for allergens suspected by history.

Author Response

Point 1: Does it really provide more useful information that just the history of symptoms consistent with allergic rhinitis? 

Response 1: We agree with the reviewer's comment that Does it really provide more useful information that just the history of symptoms consistent with allergic rhinitis? 

We have prepared a paper to detect atopic elements in nasal discharge samples from patients with AR who have serum-specific IgE and are thought to be sensitized to the antigen when they present to the clinic with nasal symptoms. When a patient is seen for nasal symptoms, even if the nasal discharge findings are not typical, the AW test can be used to search for possible atopic status in the nasal discharge. If feasible, it is preferable to be able to collect the samples in the nasal cavity.

We added this suggestin into the revised manuscript.

( Discussion p7. line 206-208 )

Point 2:  Skin testing for suspected inhalant allergens identifies allergen specific IgE rapidly and is the more usual means of evaluation. What, if any, then is there any reason to use the AW test since it provides less useful information than allergy skin tests for allergens suspected by history?

Response 2: We agree with the reviewer’s comment.

We believe that skin tests, including the prick test, are antigen-specific and rapid.

We have added a note on the advantages of performing the AW test.

That we planned to use the AW test is useful to test for comprehensive atopic status, since the suspected antigen may not be available by skin reaction with presumed antigens.

( Discussion p7. line 208-210 )

Reviewer 3 Report

Comments and Suggestions for Authors

The Article „ Development of an allergic rhinitis diagnosis application using 2 the total tear IgE detection kit for examining nasal fluid: 3 comparison and combination with the conventional nasal smear 4 examination for eosinophils” highlights the role of IgE detection in nasal fluid in the management of patients with suspicion of allergic rhinitis. I have read the paper with interest and feel that it is relevant for the area of Allergic Rhinitis management.

I suggest minor revisions: English language and style are fine/minor spell check required

Author Response

The Article „ Development of an allergic rhinitis diagnosis application using the total tear IgE detection kit for examining nasal fluid: comparison and combination with the conventional nasal smear examination for eosinophils” highlights the role of IgE detection in nasal fluid in the management of patients with suspicion of allergic rhinitis. I have read the paper with interest and feel that it is relevant for the area of Allergic Rhinitis management.

We wish to express our strong appreciation to the reviewer for these comments on our paper.

Reviewer 4 Report

Comments and Suggestions for Authors

The revised manuscript can be accepted.

Author Response

The revised manuscript can be accepted.

We wish to express our strong appreciation to the reviewer for these comments on our paper.